# Multi-Scenario Land Use Optimization Simulation and Ecosystem Service Value Estimation Based on Fine-Scale Land Survey Data

Rui Shu [1,2,*], Zhanqi Wang [1], Na Guo [3], Ming Wei [4], Yebin Zou [5] and Kun Hou [6]

1 School of Public Administration, China University of Geosciences (CUG), Wuhan 430074, China; zhqwang@cug.edu.cn
2 Ningxia Natural Resources Survey and Investigation Institute, Yinchuan 750002, China
3 Ningxia Natural Resources Information Center, Yinchuan 750002, China; gn09512024@163.com
4 Ningxia Department of Natural Resources, Yinchuan 750002, China; viman2004@sohu.com
5 School of Civil and Hydraulic Engineering, Ningxia University, Yinchuan 750021, China; zouyebin@sina.cn
6 School of Remote Sensing and Geomatics Engineering, Nanjing University of Information Science and Technology, Nanjing 210044, China; kunhou@nuist.edu.cn
* Correspondence: shurui_92@163.com; Tel.: +86-951-6981635

**Abstract:** Land optimization simulation and ecosystem service value (ESV) estimation can better serve land managers in decision-making. However, land survey data are seldom used in existing studies, and land optimization constraints fail to fully consider land planning control, and the optimization at the provincial scale is not fine enough, which leads to a disconnection between academic research and land management. We coupled ESV, gray multi-objective optimization (GMOP), and patch-generating land use simulation (PLUS) models based on authoritative data on land management to project land use and ESV change under natural development (ND), rapid economic development (RED), ecological land protection (ELP), and sustainable development (SD) scenarios in 2030. The results show that construction land expanded dramatically (by 97.96% from 2000 to 2020), which encroached on grassland and cropland. This trend will continue in the BAU scenario. Construction land, woodland, and cropland are the main types of land used for expansion, while grassland and unused land, which lack strict use control, are the main land outflow categories. From 2000 to 2030, the total amount of ESV increases steadily and slightly. The spatial distribution of ESV is significantly aggregated and the agglomeration is increasing. The policy direction and land planning are important reasons for land use changes. The land use scenarios we set up can play an important role in preventing the uncontrolled expansion of construction land, mitigating the phenomenon of ecological construction, i.e., "governance while destruction", and promoting food security. This study provides a new approach for provincial large-scale land optimization and ESV estimation based on land survey data and provides technical support for achieving sustainable land development.

**Keywords:** ecosystem service value; GMOP-PLUS coupling model; multi-scenario simulation; fine-scale land use optimization; land survey

## 1. Introduction

Ecosystem services are all the benefits that humans obtain from ecosystems [1], including food production, water conservation, climate regulation, biodiversity conservation, and other services [2,3]. Ecosystem services become a bridge between natural ecosystem functioning and human well-being and one of the most important ways for humans to perceive and evaluate the state of ecological security [4]. However, existing socio-economic systems and government performance assessments do not adequately value ecosystem assets [5,6], and ecosystem services are viewed as abundant free public services, leading to the over-consumption of ecological services, which has a serious impact on human

well-being [7–9]. Therefore, understanding and valuing ecosystem service value (ESV) can facilitate policymakers to be able to fully consider ecosystem services when balancing competing land uses and making ecologically sustainable development decisions [10–12]. The estimation methods for ESV can be grouped into two types: the primary data-based approach and the equivalent factor method for estimating land use value per unit area [13,14]. When estimating ESV on the basis of the primary data approach, it is difficult to assess and standardize the value of each type of ecosystem service using a unified approach due to the difficulty of ecological process model construction, multiple model input parameters, and complex simulation processes [15,16]. Therefore, this approach is usually applied to small spatial scales or single ecosystems [17,18]. This paper uses a modified equivalence factor (see Section 3.2 for details) to calculate ESV, which can evaluate the spatiotemporal changes of large-scale and diverse ecosystem services [15,19].

Land use optimization includes quantity optimization and spatial optimization. Quantitative optimization models mainly include the gray model (GM), the system dynamics (SD) model [14], gray multi-objective optimization (GMOP) [20], Markov [21], etc. The GM model considers a single factor, and the long-term prediction results are unreliable. The selection of factors and the description of the long-range, comprehensive, and trend nature of SD model construction is subjective and difficult to accurately quantify [22]. The Markov modeling process requires land use data to be stable and is not applicable to areas that are subject to complex factors over long periods of time [23,24]. GMOP has been widely used because it can fully consider the uncertainty of future land use and utilize multiple constraints to solve the objective problem [19,25,26]. The main spatial optimization models are the conversion of land use and its effects at small region extent (CLUE-S) model [19,27], agent-based (ABS) model [28,29], and cellular automata (CA) model. The CLUE-S model's assumptions about causality are not always fully justified. At the same time, the model does not have a process mechanism to correct the system parameters for some land use change processes. The ABS model focuses on local scales and cannot be directly applied to large scales and high resolutions. Meanwhile, the simulation process is too mechanized, and the parameters are set randomly, which cannot reflect the real-world network structure [30]. CA models can predict complex land use processes by setting simple rules and have been widely applied at different scales [31–33]. Currently, the coupled CA model that takes into account both quantity and space has become a hot topic in the study of land use pattern optimization [34]. Common CA models include CA-Markov [23], SD-CA [14], artificial neural network (ANN)-CA [35], future land use simulation (FLUS) [36], patch-generating land use simulation (PLUS) [37], etc. Among them, PLUS is based on the random forest algorithm and adaptive inertia competition mechanism, which can better mine all kinds of land use change triggers. Combined with random seed generation and a threshold-decreasing mechanism, it can realize the dynamic simulation of land use change at the patch level. It has been widely proven to have higher accuracy in land use simulation [22,38].

The constraint mechanism is a key component of land use simulations [34]. At present, the Chinese government's control over land is mainly reflected in the rigid constraints of the Territorial Spatial Plan (2021–2035) (TSP) (Figure 1). Most of the current studies have two drawbacks for spatial constraints on land optimization. Firstly, some of the studies incorporated the red lines for protecting ecosystems (RLE) directly into the prohibited conversion zone [19,22], which is obviously unreasonable. Although construction and farming are not allowed within the RLE, the conversion of woodland, grassland, and unused land cannot be avoided. Secondly, little research has included permanent basic cropland (PBC) and boundaries for urban development (BUD) in spatial constraints [20,25,39]. As for the management of land, construction projects must be within the BUD (except for nationally important infrastructure projects), and PBC is strictly prohibited from being occupied by construction and planted with non-food crops.

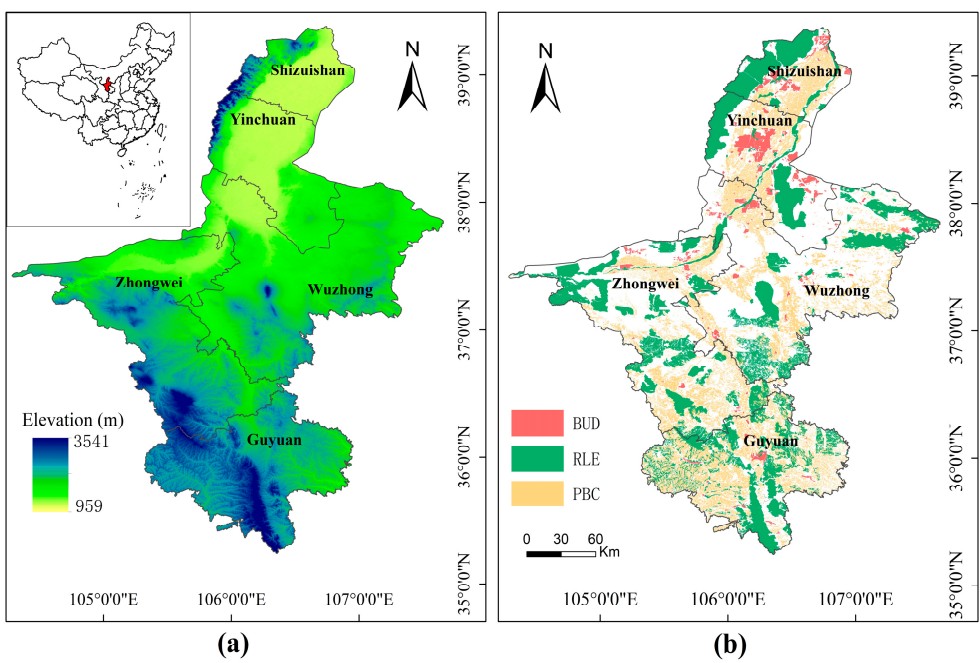

**(a)**  **(b)**

PBC, RLE, and BUD are the boundary controls of the TSP, which puts mandatory constraint requirements on the national land space. It is an insurmountable red line for China to adjust its economic structure, plan industrial development, and promote urbanization.

PBC: High-quality arable land determined in accordance with the demand for agricultural products by the population and socio-economic development over a certain period. It may not be occupied or changed in use without authorization.

RLE: Land, water, and sea areas that have special important ecological functions and must be strictly protected on a mandatory basis. They are the bottom lines and lifelines for safeguarding and maintaining national ecological security.

BUD: This is to guide and constrain urban development for a certain period. No centralized urban construction, planning, or building of any kind in development zones and industrial parks shall be carried out outside its boundaries.

**Figure 1.** Location and basic information of the study area. (**a**) Location and DEM of Ningxia province; (**b**) national territory spatial planning (2021–2035).

The bridge between academic research and natural resource management decisions cannot be built without using land survey data [12]. However, due to the difficulties in obtaining land survey data, the lack of historical data, and inconsistent data standards, most of the existing studies use remote sensing image interpretation to obtain land use data when calculating the ESV and optimizing land [37,40,41]. Nevertheless, the accuracy of remote sensing interpretation data lacks validation and cannot meet the actual land management needs. Taking the 30 m land use data from the Resource and Environment Sciences and Data Center (https://www.resdc.cn) (accessed on 30 October 2022), commonly used in previous studies as an example [14,25,36,42], the accuracy of the area is only 69%, compared with the land survey data of Ningxia in 2020 (Table S1). In addition, it is difficult to balance the resolution and study scale in existing studies. Large-scale studies are often carried out at low resolutions, such as Wang et al. [43], Gu et al. [44], and Li et al. [45], who used 200 m to 1 km resolution data for provincial and watershed studies. Fine resolution (30 m) is often difficult to apply at provincial and higher scales. Most of the studies carried out using 30 m resolution data are concentrated at the county and municipal scales [14,39,46]. The minimum patch size for the land survey has been increased from 1500 m$^2$ to 400 m$^2$. Land use data below 30 m cannot meet the needs of land survey and management if used.

Numerous studies have shown that land use change has a significant effect on the ESV and can improve ecosystem service capacity by improving and optimizing land use

structure [25,47,48]. Therefore, the ESV can be used to assess, compare, and select land use options under multiple planning target scenarios.

The aim of our study is to optimize land use patterns from the perspective of actual land management using large-scale survey data and coupled land use models. Our objectives were the following: (1) to analyze the spatial and temporal evolution of land use and the ESV in Ningxia over the past 20 years; (2) to predict the trends of land use and ESV changes in 2030 under natural development (ND), rapid economic development (RED), ecological land protection (ELP), and sustainable development (SD) scenarios; (3) to analyze the comprehensive effect of land use changes on the ESV. Our study provides new insights for optimizing land allocation and improving land resource use efficiency in the future. It provides a basis for promoting regional ecosystem conservation.

## 2. Study Area and Data Sources

### 2.1. Study Area

Ningxia is located in the middle and upper reaches of the Yellow River and at the intersection of the desert and the Loess Plateau. It consists of five cities and covers an area of about 51,900 km$^2$ between latitudes of $35°14'$–$39°23'$ N and longitudes of $104°17'$–$107°39'$ E (Figure 1). The area has an average altitude of 1100 m, an average annual temperature of 8 °C, an average annual rainfall of less than 200 mm, and an average annual water evaporation of 1250 mm, which is typical of a temperate continental climate. Ningxia bears the important mission of maintaining ecological security in Northwest China and even the whole country, and its ecological status is of great significance. The Ningxia TSP was approved in August 2023 by the State Council, requiring that by 2035, the area of PBC in Ningxia will be no less than 9493 km$^2$, the area of the RLE no less than 12,000 km$^2$, and the expansion of the BUD no more than 1.3 times the scale of urban construction land in 2020.

### 2.2. Data and Processing

Land survey data are the basis for all land management and the most accurate and the only legal data for land cover. Ningxia has conducted three land surveys in 1986–1995, 2007–2009, and 2017–2020. On the foundation of a comprehensive survey every 10 years, an updated survey is conducted annually to ensure the timeliness of the data. The process of land survey is as follows: firstly, satellite images are used to outline the scope of land use, then field reviews are conducted to determine the types of land use, and finally, verification is conducted at the county, municipal, provincial, and national levels, patch by patch. Taking the third land survey of Ningxia as an example, there are about 2.7 million patches, and the number of patches evidenced by field photographs is as high as 1.26 million, and the number of evidenced photographs is about 11 million, which ensures the accuracy of the data. As data accuracy improves, land survey data become less accessible. The land survey data and most of the driving factors used in this study were obtained from the Ningxia Department of Natural Resources. The total amount of data is more than 500 GB, which ensures the accuracy and applicability of the research results.

Firstly, the first and second land survey data are standardized with the third land survey data. The standardization process mainly includes the normalization of land class names (Table S2) and the conversion of point and line features into surface features. Then the land use types are divided into six categories: cropland, woodland, grassland, construction land, water area, and unused land (Figure 2, Table S2). Finally, we used high-performance server clusters to convert the data into grid cells with a spatial resolution of 30 m and an extent of 9923 × 15,371. We also derived raster maps of multiple driving factors and spatial constraints with the same extent as the land use data. Table 1 lists the data source pre-processing process for this study.

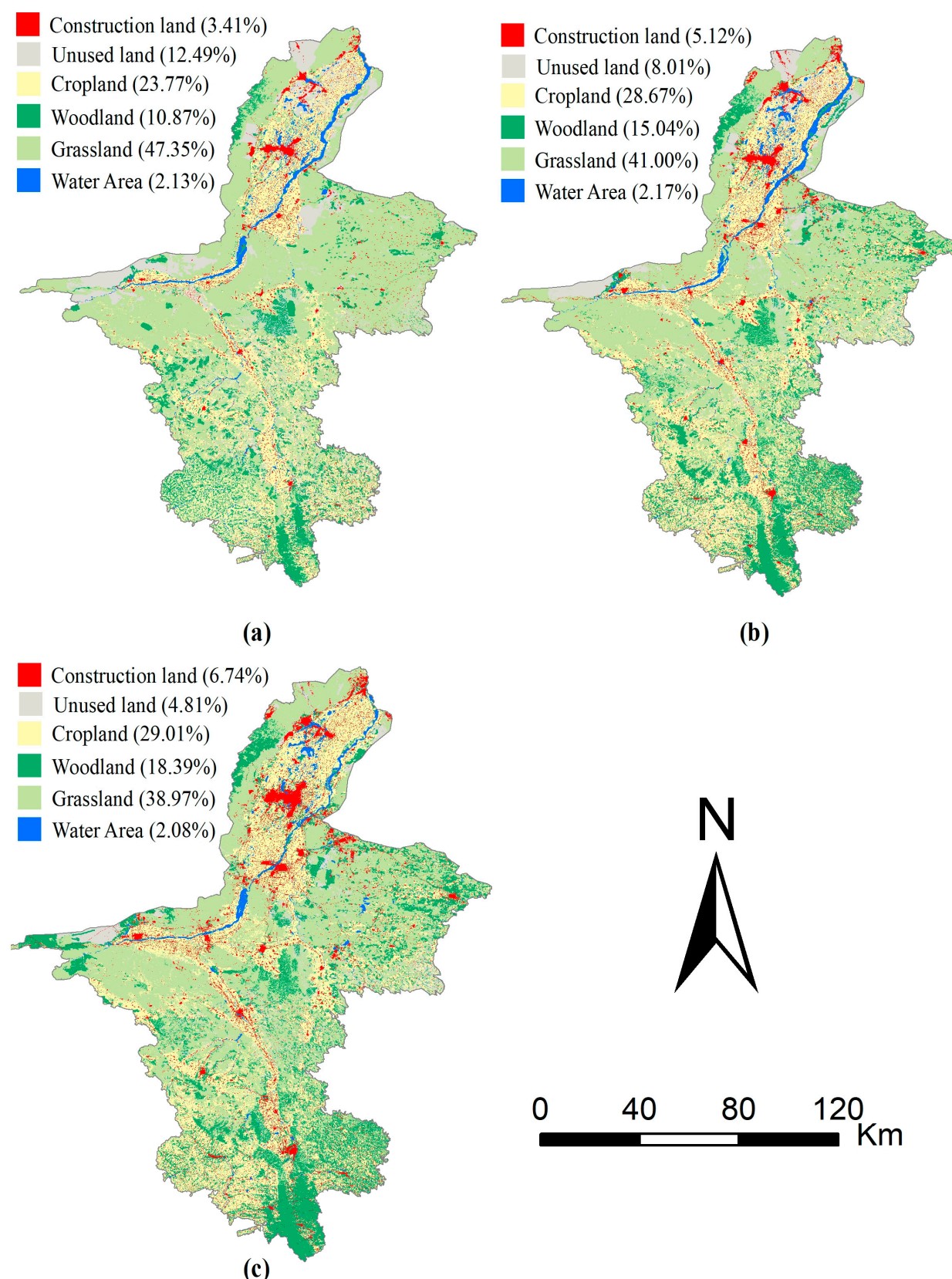

**Figure 2.** Land use map of Ningxia in (**a**) 2000, (**b**) 2010, and (**c**) 2020.

**Table 1.** Data sources and processing.

| Category | Data Name | Year | Resolution | Data Sources and Data Processing |
|---|---|---|---|---|
| Land use | Land survey data | 2000, 2010, 2020 | Vector | Ningxia Department of Natural Resources. |
| Socio-economic statistics data | Population, GDP, Grain yield, Planting area, Food prices, Output value of agriculture, forestry, animal husbandry, and fishery | 2010–2020 | Non-spatial data | Ningxia Statistical Yearbook 2010–2020 (https://tj.nx.gov.cn/) (accessed on 20 October 2022) Ningxia Grain and Material Reserve Bureau (http://lswz.nx.gov.cn/) (accessed on 15 March2023) |
| Socio-economic spatial data | Population density | 2019 | 1 km | Resource and Environmental Science and Data Center (http://www.resdc.cn) (accessed on 30 October 2022) |
| | GDP | 2019 | 1 km | |
| | Nighttime light intensity | 2020 | 0.004° | |
| Climate and environmental data | Average annual precipitation, Average annual ground temperature, Average annual evaporation, Soil type | 1960–2010 | 1 km | Resource and Environment Sciences and Data Center (http://www.resdc.cn) (accessed on 30 October 2022) |
| Spatial accessibility data | Distance to rural settlements | 2020 | 30 m | The range of settlements and towns was extracted from the 2020 Land Change Survey data. The range of development zones was extracted from the 2020 Land Intensification Evaluation data of development zones. The data on roads and rivers were extracted from geographic state monitoring. The distances to rural settlements, towns, open economic zones, major rivers, railroads, national roads, provincial roads, and other roads are calculated in ArcGIS with the "near" tool. |
| | Distance to town | 2020 | 30 m | |
| | Distance to open economic zone | 2020 | 30 m | |
| | Distance to major rivers | 2020 | 30 m | |
| | Distance to railroads, national roads, provincial roads, and other roads | 2020 | 30 m | |
| Spatial constraints data | Permanent basic farmland | 2022 | Vector | Ningxia Territorial Spatial Planning (2021–2035) |
| | Urban development boundary | 2022 | Vector | |
| | Key projects | 2021–2035 | Vector | Outline of the 14th Five-Year Plan and Vision 2035 |

## 3. Methods

### 3.1. The GMOP-PLUS Model

We proposed a method for coupling the GMOP and the PLUS models. The coupled model consists of two steps. Firstly, the GMOP algorithm is used to obtain optimal land use solutions under different scenarios by inputting decision variables, constraints, and objective functions. Secondly, parameters such as land change drivers, land use conversion rules, domain weights, and spatial restrictions are input into the PLUS model to realize the spatialized representation of land use quantity changes (Figure 3).

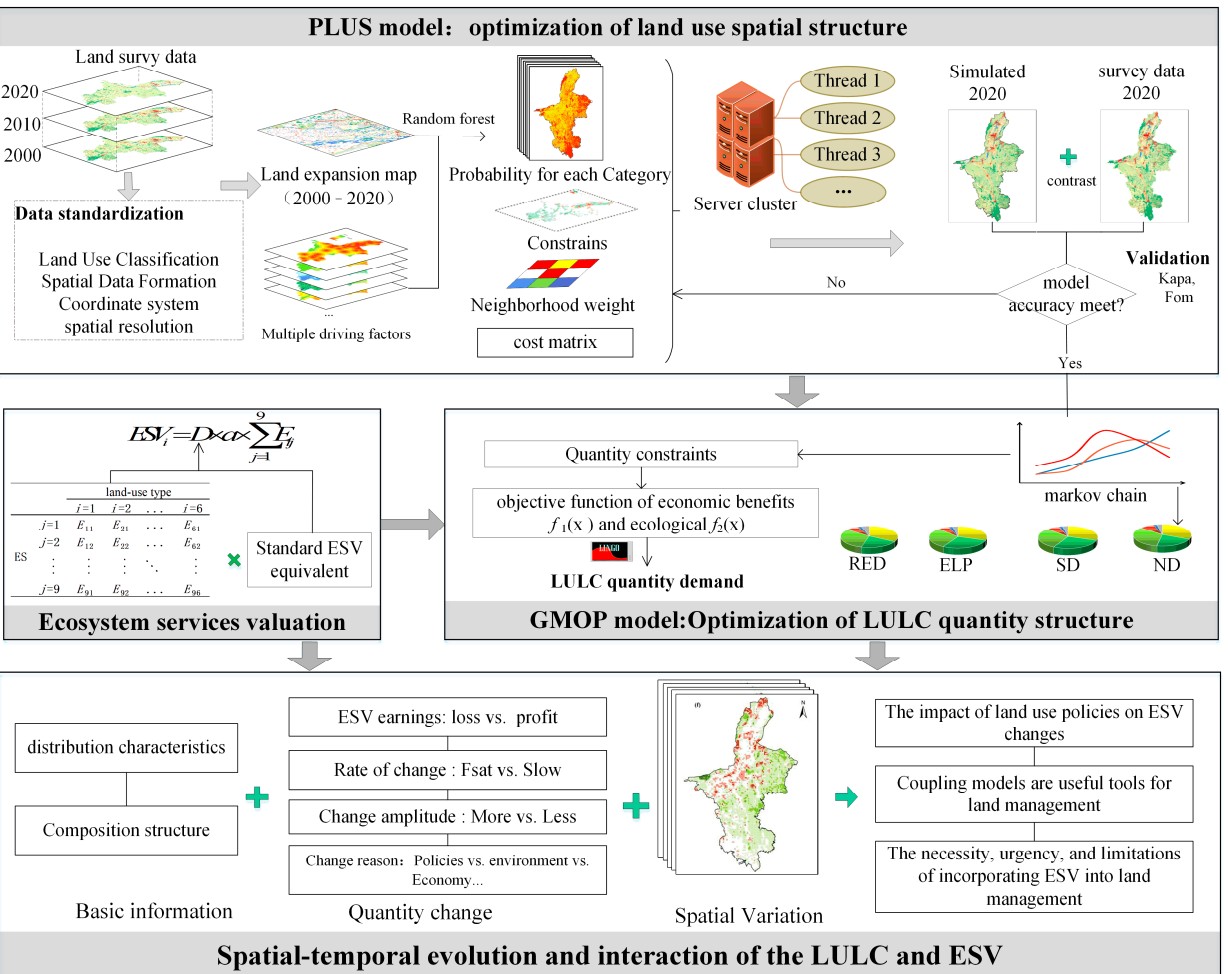

**Figure 3.** Flow chart of land utilization simulation and ESV evaluation.

### 3.1.1. Optimization of Land Use Structure Using GMOP Algorithm

- Scenario Setting

We predicted the areas of each land use type in 2030 under four scenarios in Ningxia using Lingo 12.0 software.

Natural development scenario: The scenario follows the natural evolution of land without considering any policy constraints and spatial controls. It is obtained using Markov chain projections based on land use data for 2010 and 2020.

Rapid economic development scenario: The scenario maximizes the economic benefits of land use.

$$f_1(x) = max\sum_{i=1}^{6} x_i d_i \tag{1}$$

where $f_1(x)$ is the maximum economic output of land use. $x_i$ is the total area of the i-th land use type. $d_i$ is the economic output value per unit area of the *i*-th land use type. The economic benefits of cropland, woodland, grassland, and water were estimated by the output value of plantation, forestry, animal husbandry, and fishery, respectively. The GDP of secondary and tertiary industries was used to estimate the economic benefits of construction land. We collected historical data from the Ningxia Statistical Yearbook (2010–2020) and uniformly imputed to comparable 2010 prices. The economic production value of cropland, woodland, grassland, construction land, and water area in 2030 was predicted to be 2.84, 0.04, 1.23, 149.92, and 3.10 (unit: $10^4$CNY/ha) using the GM (1, 1) model, respectively.

Ecological land protection scenario: The scenario maximizes the ecological value of land use.

$$f_2(x) = max \sum_{i=1}^{6} x_i e_i \tag{2}$$

where $f_2(x)$ is the maximum ESV of land use, and $e_i$ is the ESV of the ith land use type. The ESV of cropland, woodland, grassland, construction land, water area, and unused land are 1.84, 6.54, 2.72, $-1.32$, 11.05, and 0.32 (unit: $10^4$CNY/ha), respectively (see Section 3.2 for details).

Sustainable development scenario: To achieve harmonious economic and ecological development, it is necessary to maximize the ecological and economic value of land use.

$$max\{f_1(x), f_2(x)\} \tag{3}$$

- Constraints

(1) The RED, ELP, and SD scenarios are also subject to the conditions of historical land use, TSP control, strategic objectives, and development vision.

$$\sum_{i=1}^{7} x_i = 5,196,400 \text{ hm}^2 \tag{4}$$

(2) Constraint of the total population. The total population should be below the carrying capacity of land resources for human activities. Based on the population and land statistics from 2010 to 2020 and the GM (1,1) algorithm, the total population of Ningxia is predicted to reach 8.19 million, and the average population densities of construction land and agricultural land (woodland, grassland, and cropland) will be 16.38 and 0.42 people per hectare by 2030, respectively.

$$0.42 \times (x_1 + x_2 + x_3) + 16.38 \times x_4 \leq 8.19 \times 10^6 \tag{5}$$

(3) Constraint of food security.

$$\otimes \alpha_{31} \times f_0 \times f_1 \times 0.81 x_1 \geq s \times w \times p \tag{6}$$

where $\alpha_{31}$ is the grain yield per unit area, its lower limit is the current grain yield, and the upper limit is the grain yield in the target year. $f_0$ is the average number of crops planted per year on the same cropland. $f_1$ is the proportion of food crops grown on cropland. $s$ is the per capita food demand, which is expected to reach 517.3 kg/a in 2030 [49]. w is the grain self-sufficiency rate. Ningxia has the second largest arable land per capita and is one of the twelve commercial grain production bases in China. Grain production should at least meet the requirement of self-sufficiency, so w is taken as 100%. p is the population size in the forecast year. Based on the GM (1, 1) model and historical data, we predict that $f_0 = 0.97$, $f_1 = 0.49$, and $\otimes \alpha_{31} \in (5603.68, 6931.77)$. According to historical land use data, after removing supporting facilities such as ditches and roads, the area of cropland that can really be used for cultivation is about 81%.

(4) Constraint of cropland area. The TSP sets the target of cropland protection for Ningxia at 1,169,220 hm². According to the "Guidelines for the Evaluation of the Suitability of Resource and Environment Carrying Capacity and Territorial Spatial Development ", under the constraints of land resources and water resources, the upper limit of the cropland carrying capacity in Ningxia is 1,406,310 hm².

$$1,169,220 \text{ hm}^2 \leq 0.81 x_1 \leq 1,406,310 \text{ hm}^2 \tag{7}$$

(5) Constraint of woodland area. Ningxia's forest coverage rate rose from 11.9% in 2012 to 15.8% in 2020. According to the target of Ningxia's 14th Five-Year Plan (2021–2025), the

forest coverage rate should reach 20% in 2025. Obviously, the growth rate of the woodland area will be higher than in the ND scenario.

$$1,080,925 \text{ hm}^2 \leq x_2 \tag{8}$$

(6) Constraint of construction land area. Construction land includes urban construction land, village construction land, and other construction land. According to the TSP, the area of urban construction land in 2030 will not exceed 1.2 times that of in 2020. With the decrease in village population, the village construction land in 2030 will remain unchanged. Other construction land in 2030 will be calculated based on the average growth rate from 2010 to 2020. The lower limit of construction land in 2030 is the construction land area in 2020.

$$350,487 \text{ hm}^2 \leq x_4 \leq 398,390 \text{ hm}^2 \tag{9}$$

(7) Constraint of grassland area. In 2030, the probability is that grassland areas will remain on a downward trend. Therefore, the upper limit of the grassland area is the current grassland area, and the lower limit is the grassland area with 10% downward of the ND scenario.

$$1,549,577 \text{ hm}^2 \leq x_3 \leq 2,024,176 \text{ hm}^2 \tag{10}$$

(8) Constraint of water area. Ningxia's water area has remained stable over the years. Therefore, the minimum and maximum water areas from 2000 to 2020 are adopted as the bound area of the watershed in 2030.

$$96,700 \text{ hm}^2 \leq x_6 \leq 110,011 \text{ hm}^2 \tag{11}$$

(9) Constraint of unused land area. Under the double pressure of strict land use control measures and economic development, a large amount of unused land can only be reclaimed to meet the development. Hence, the upper limit of the unused land area is the currently existing unused land, and the lower limit is the area of unused land under the ND scenario.

$$165,791 \text{ hm}^2 \leq x_7 \leq 249,720 \text{ hm}^2 \tag{12}$$

3.1.2. Optimization of Land Use Spatial Allocation by PLUS Model

The PLUS model includes two parts: a rule mining framework for the Land Expansion Analysis Strategy (LEAS) and a CA model based on multi-class random patch seeding (CARS) in two parts. The former was randomly sampled from the portion of each type of land use increase between the two periods of land use change. Then the random forest algorithm is used to mine each type of land use expansion and the driving factors one by one. Finally, the development probability of each type of land use and the degree of contribution of the driving factors to the expansion of each type of land use in that period are obtained. The latter is based on the adaptive inertia competition mechanism of roulette to obtain the overall probability of land use change. The final land use approach is then achieved through multiple iterations by combining random patch generation and the threshold decreasing mechanism. The specific parameters are set as follows:

- Spatial driving factors

We referred to previous research [19,25], and while considering the principles of representativeness, the quantifiability of factors, and the availability of information, 19 driving factors were selected for our study (Table 1, Figure 4).

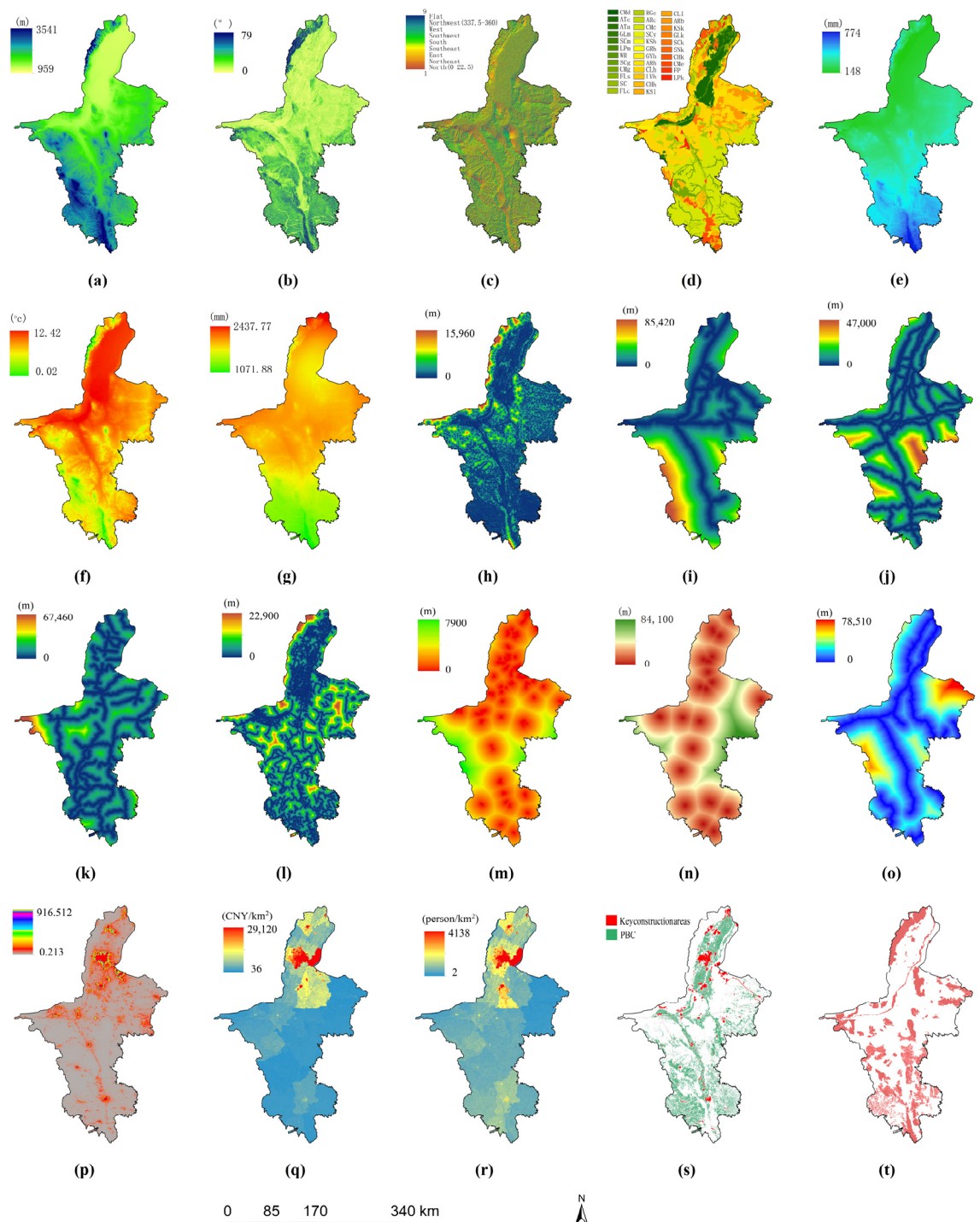

**Figure 4.** The spatial driving factors and spatial restrictions of the land use change in this study. (**a**) Dem; (**b**) Slope; (**c**) Aspect; (**d**) soil type e; (**e**) Precipitation; (**f**) Temperature; (**g**) Evaporation; (**h**) Dis rural settlements; (**i**). Dis railroads; (**j**) Dis national road; (**k**) Dis provincial road; (**l**) Dis other road; (**m**) Dis open economic zone; (**n**) Dis town; (**o**) Dis main rivers; (**p**) Nighttime light intensity; (**q**) GDP; (**r**) Population density; (**s**) Spatial restrictions; (**t**) RLE.

- Spatial restrictions

  We set the PBC as the prohibited conversion zone and the BUD and the planned key projects as the priority development zones of the construction land.

- Conversion elastic coefficient and conversion matrix setting

The elasticity coefficient of land use conversion is the degree of difficulty in converting a certain type of land use to another type. It can be defined by the model parameter ELAS (taking values from 0 to 1), with higher values indicating higher stability of land use types and a lower probability of conversion. In our research, the ELAS for the ND scenario is determined using the rate of land use change from 2000 to 2020, and the values of RED, ELP, and SD are determined based on the specifics of each scenario and other studies (Table S3).

The cost matrix between different land use types represents whether they can be transferred to each other. The ND scenario is unrestricted and can be converted between different land use types. The RED scenario does not allow the conversion of construction land into other land use types. Under RED, ELP, and SD scenarios, cropland and construction land cannot be degraded into water areas and unused land (Table S4).

- Model validation

Fom and Kapa are often used to assess the accuracy and consistency of simulation results. Fom reflects the accuracy of the simulation results by considering only the number of cells that change during the simulation and excluding the invariant cells that lead to erroneous accuracy [50]. Kappa is calculated based on the confusion matrix, reflecting the consistency of the simulation results [51]. Both Fom and Kappa take values ranging from 0 to 1, with larger values indicating a higher simulation accuracy. However, Fom values are usually less than 0.3 [22,37]. When Fom is close to 0.3 and the Kappa coefficient is greater than 0.6, it indicates that the model simulation accuracy achieves better results in a statistically significant way.

*3.2. Ecosystem Services*

3.2.1. Ecosystem Services Valuation

Wang et al. [52] revised the model proposed by Costanza et al. [53] and developed the equivalent factor table of ecosystem service values in China (Table S5). However, the model set the ESV of construction land to 0 [19,42]. In fact, construction land can provide recreation and cultural services and have negative ecological impacts due to pollution [48]. In our study, we made two modifications to the model proposed by Xie [54]. First, the unused land is dominated by bare land and sandy land, so it corresponds to desert [55]. Water area is the area-weighted average of the water system and wetland [44]. Second, the negative impacts of construction land on ecosystem services are estimated using a proxy cost approach, in which water resource conservation is approximately calculated based on the annual average value of domestic and commercial water use, and waste treatment is estimated based on the social labor value consumed in treating the three wastes [56]. The ESV in each grid is calculated as follows:

$$ESV_i = D \times a \times \sum_{j=1}^{9} E_{ij} \tag{13}$$

$ESV_i$ is the ESV for grid with land use type *i*. D is a standard ecosystem service value equivalent, equal to 1/7 of the market value of grain yield per hectare. D is calculated from the yield and area of major grain crops (corn, wheat, rice, soybean) in Ningxia from 2010 to 2020, and predicted using the GM (1, 1) model to obtain D in 2030 as 2327.5 CNY/ha. a is the area of each grid, equal to 0.9 ha. $E_{ij}$ is the value equivalent of the j-th ecosystem service for a grid with land use type *i*.

3.2.2. Spatial Autocorrelation Analysis

The global Moran index is used to determine whether the ESVs in the study area are spatially clustered and the degree of clustering. The calculation formula is as follows:

$$moran's = \frac{n\sum_{i=1}^{n}\sum_{j=1}^{n} w_{ij}(x_i - \bar{x})(x_j - \bar{x})}{\sum_{i=1}^{n}\sum_{j=1}^{n} w_{ij}\sum_{i=1}^{n}(x_i - \bar{x})^2} \tag{14}$$

where $n$ is the total number of spatial cells, $x_i$ and $x_j$ represent the ESV of $i$ and $j$ spatial cells, $\bar{x}$ is the average of the ESV of all spatial cells, $w_{ij}$ is the spatial weight value. If $i$ is greater than 0, it means that regions with large (small) ESV values are more likely to cluster together.

## 4. Results and Analysis

We simulated the land use in 2020 using the land use data from 2000 and 2010. Then the simulated result was compared with the actual land use in 2020 (Figure S1). The results showed that the Fom is 0.2526 and the Kappa is 0.7442, which indicates that the simulation accuracy of this experiment is high, and the model parameters are set reasonably.

### 4.1. Spatial and Temporal Variation Characteristics of Land Use

#### 4.1.1. Variation in Land Use between 2000 and 2020

As shown in Table 2 and Figure 5, the structure and distribution of land use in Ningxia have undergone significant changes. From 2000 to 2020, the area of grassland decreased by 4355 km$^2$, which is the land type with the largest area change. The expansion of construction land has nearly doubled, with a continuous increase in area from 1770 km$^2$ to 3505 km$^2$, with the expansion mainly occupying cropland (43.95%), grassland (34.99%), and unused land (11.74%). Similarly, the area of woodland increased by 69.2% over 20 years, from 5646 km$^2$ in 2000 to 9555 km$^2$ in 2020. A total of 60.73% of the growth area of woodland that originated from grassland and 24.09% from cropland returned to forest. Unused land decreased by 61.5%, with the reduced area mainly flowing to grassland (55.53%) and farmland (20.03%). The water area remained basically unchanged. It is noteworthy that the land change between 2000 and 2010 is more drastic than that between 2010 and 2020.

#### 4.1.2. Projection of Land Use Changes in 2030 Based on GMOP-PLUS Model

To explore the impact of the PBC, RLE, and BUD on land optimization, four sample plots were selected to explore and compare the land use structure (Figure 6). As shown in Table 2, all four scenarios show a steady increase in farmland area, with all sources of increase mainly coming from grassland. The area of construction land remains undiminished. As expected, compared to 2020, the ND scenario has the highest expansion rate of construction land (18.57%), much higher than the RED scenario (10.10%) and the SD scenario (9.47%). It remains largely unchanged in the ELP scenario. Woodland shows a more substantial increase, with a 13.12% increase under the ND scenario and RED scenario, and a nearly 50% increase under both the ELP scenario and SD scenario. The water combines high economic and ecological benefits, coupled with the strict national control of water consumption on the Yellow River (90% of Ningxia's water resources originate from the Yellow River), and the RED, ELP, and EEB scenarios all show a small increase in the water area. Land change tends to seek to maximize economic benefits. Therefore, the ND scenario and RED scenario show a negative growth of ecological land. In contrast, the ELP scenario and SD scenario emphasize the importance of ecology, and the ecological land has a growth trend.

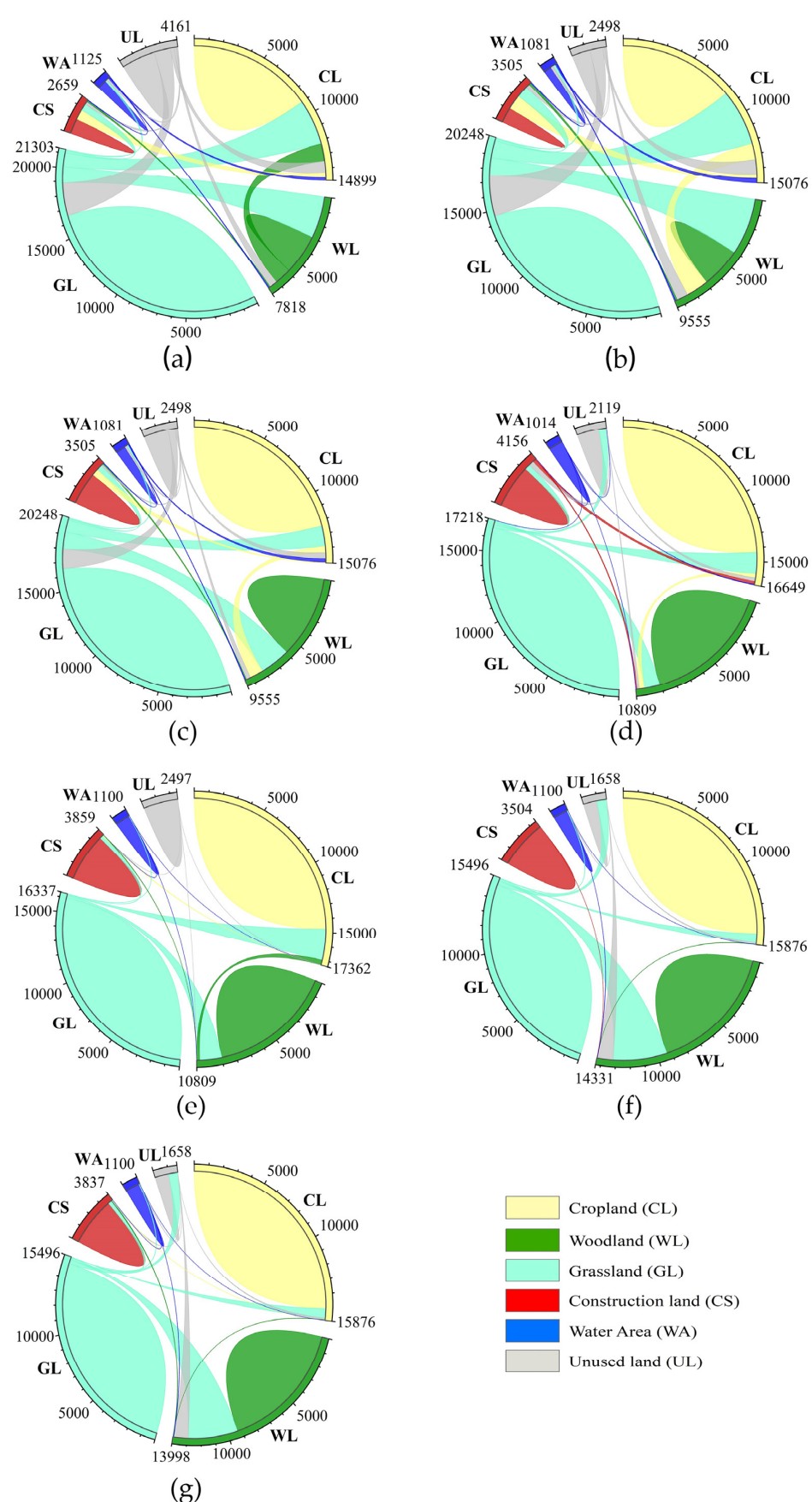

**Figure 5.** Land use transfer in different periods: (**a**) 2000–2010; (**b**) 2010–2020; (**c**) 2000–2020; (**d**) 2020–ND; (**e**) 2020–RED; (**f**) 2020–ELP; (**g**) 2020–SD.

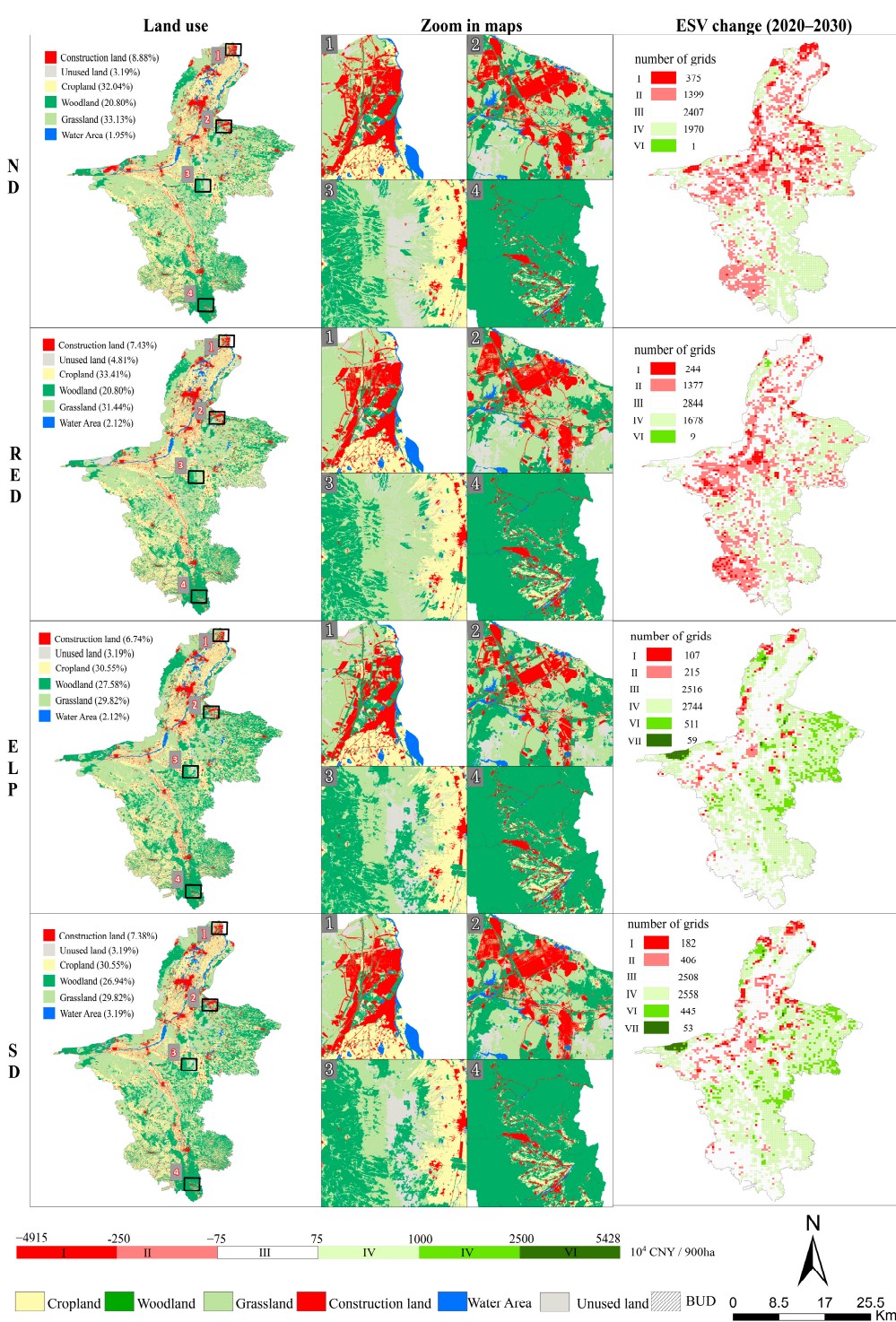

**Figure 6.** Comparison of simulation land use structure and the ESV changes in 2030 under different scenarios.

**Table 2.** Variations in land use from 2000 to 2030.

| LU Type | Land Use (km²) | | | | | | | Relative Change Rate (%) | | | | | | |
|---|---|---|---|---|---|---|---|---|---|---|---|---|---|---|
| | 2000 | 2010 | 2020 | ND | RED | ELP | SD | 2000–2010 | 2010–2020 | 2000–2020 | 2020–ND | 2020–RED | 2020–ELP | 2020–SD |
| Ecological land | 43,704 | 45,144 | 45,961 | 45,690 | 45,608 | 46,801 | 46,469 | 3.29 | 1.81 | 5.16 | −0.59 | −0.77 | 1.83 | 1.11 |
| Cropland | 12,350 | 14,899 | 15,076 | 16,649 | 17,362 | 15,876 | 15,876 | 20.64 | 1.19 | 22.07 | 10.43 | 15.16 | 5.30 | 5.30 |
| Woodland | 5646 | 7817 | 9555 | 10,809 | 10,809 | 14,330 | 13,998 | 38.46 | 22.23 | 69.24 | 13.12 | 13.12 | 49.97 | 46.50 |
| Grassland | 24,603 | 21,303 | 20,248 | 17,218 | 16,337 | 15,496 | 15,496 | −13.41 | −4.95 | −17.70 | −14.97 | −19.32 | −23.47 | −23.47 |
| Construction land | 1770 | 2659 | 3505 | 4156 | 3859 | 3505 | 3837 | 50.16 | 31.83 | 97.96 | 18.57 | 10.10 | 0.00 | 9.47 |
| Water area | 1105 | 1125 | 1081 | 1014 | 1100 | 1100 | 1100 | 1.79 | −3.93 | −2.20 | −6.20 | 1.77 | 1.77 | 1.77 |
| Unused land | 6489 | 4161 | 2498 | 2119 | 2497 | 1658 | 1658 | −35.87 | −39.96 | −61.50 | −15.20 | −0.05 | −33.64 | −33.64 |

## 4.2. Variations in Ecosystem Service Value during 2000–2030

### 4.2.1. Temporal Estimation in Ecosystem Service Value

As shown in Table 3, in general, the proportion of the ESV of each function to the total ESV did not change much. Ecosystem services in the study area mainly consist of regulating services and supporting services, which account for 50.8% and 31.6% of the total ESV in 2020, respectively. From 2000 to 2020, the total ESV in Ningxia increased by 14.9 billion CNY, with a growth rate of 10.74%. Specifically, all types of ecosystem services increase except waste treatment, which decreases, with the growth rate of raw material (35.24%) much higher than other ecosystem services. From 2020 to 2030, the total ESV continues to show an increasing trend for all scenarios. Each individual ESV is also increasing, except for water conservation and waste treatment in the ND scenario and waste treatment and soil formation and protection in the RED scenario. Undoubtedly, the ELP scenario has the largest increase in the total ESV, while the ND scenario and RED scenario have the lowest increase in the total ESV at 0.78% and 1.03%, much lower than the ELP scenario (12.91%) and the SD scenario (11.21%).

**Table 3.** Variations in ecosystem service value during 2000–2030.

| Primary Classification | Secondary Classification | ESV (10⁹ CNY) | | | | | | | ESV Relative Changes (%) | | | | |
|---|---|---|---|---|---|---|---|---|---|---|---|---|---|
| | | 2000 | 2010 | 2020 | ND | RED | ELP | SD | 2000–2020 | 2020–ND | 2020–RED | 2020–ELP | 2020–SD |
| Supply services | Food production | 59.27 | 63.48 | 64.03 | 65.54 | 66.44 | 64.80 | 64.55 | 8.02 | 2.37 | 3.76 | 1.21 | 0.81 |
| | Raw material | 72.43 | 86.83 | 97.95 | 105.48 | 105.49 | 127.77 | 125.47 | 35.24 | 7.69 | 7.70 | 30.45 | 28.10 |
| Regulating service | Air regulation | 166.68 | 180.98 | 194.68 | 199.20 | 197.57 | 227.42 | 224.08 | 16.80 | 2.32 | 1.48 | 16.82 | 15.10 |
| | Climate regulation | 184.61 | 198.46 | 210.45 | 214.10 | 213.56 | 240.25 | 237.10 | 14.00 | 1.73 | 1.48 | 14.16 | 12.66 |
| | Water conservation | 194.85 | 201.65 | 205.79 | 201.82 | 205.97 | 236.57 | 230.73 | 5.62 | −1.93 | 0.09 | 14.96 | 12.12 |
| | Waste treatment | 169.92 | 170.93 | 167.81 | 162.42 | 166.90 | 175.12 | 171.90 | −1.24 | −3.21 | −0.54 | 4.35 | 2.43 |
| Supporting services | Soil formation and protection | 227.89 | 238.83 | 249.39 | 250.51 | 248.66 | 271.80 | 268.69 | 9.44 | 0.45 | −0.29 | 8.99 | 7.74 |
| | Biodiversity conservation | 210.68 | 223.16 | 235.25 | 238.13 | 237.05 | 266.03 | 262.55 | 11.66 | 1.22 | 0.76 | 13.08 | 11.60 |
| Cultural services | Recreation and culture | 98.22 | 102.46 | 107.86 | 107.90 | 107.34 | 121.43 | 120.01 | 9.81 | 0.04 | −0.48 | 12.58 | 11.27 |
| | total | 1384.55 | 1466.79 | 1533.22 | 1545.10 | 1548.98 | 1731.19 | 1705.08 | 10.74 | 0.78 | 1.03 | 12.91 | 11.21 |

### 4.2.2. Spatial Characteristics of Ecosystem Service Value

To further analyze the spatial distribution of the ESVs and their degree of change in the study area, we created a 3 km × 3 km square grid, and then calculated the ESV and change values for each grid, and finally produced a spatial distribution (Figure 7) and change map of the ESVs from 2000–2030 (Figure 6). The ESV in the study area is dominated by medium values and medium-low values. From 2000 to 2020, the area with ESVs of low value, medium-low value, and medium value has been decreasing, and the area of medium-high value and high value has been increasing. This trend will continue under the ELP scenario and SD scenario. However, in the ND scenario and RED scenario, except for a decrease in the median value area, all other areas increased.

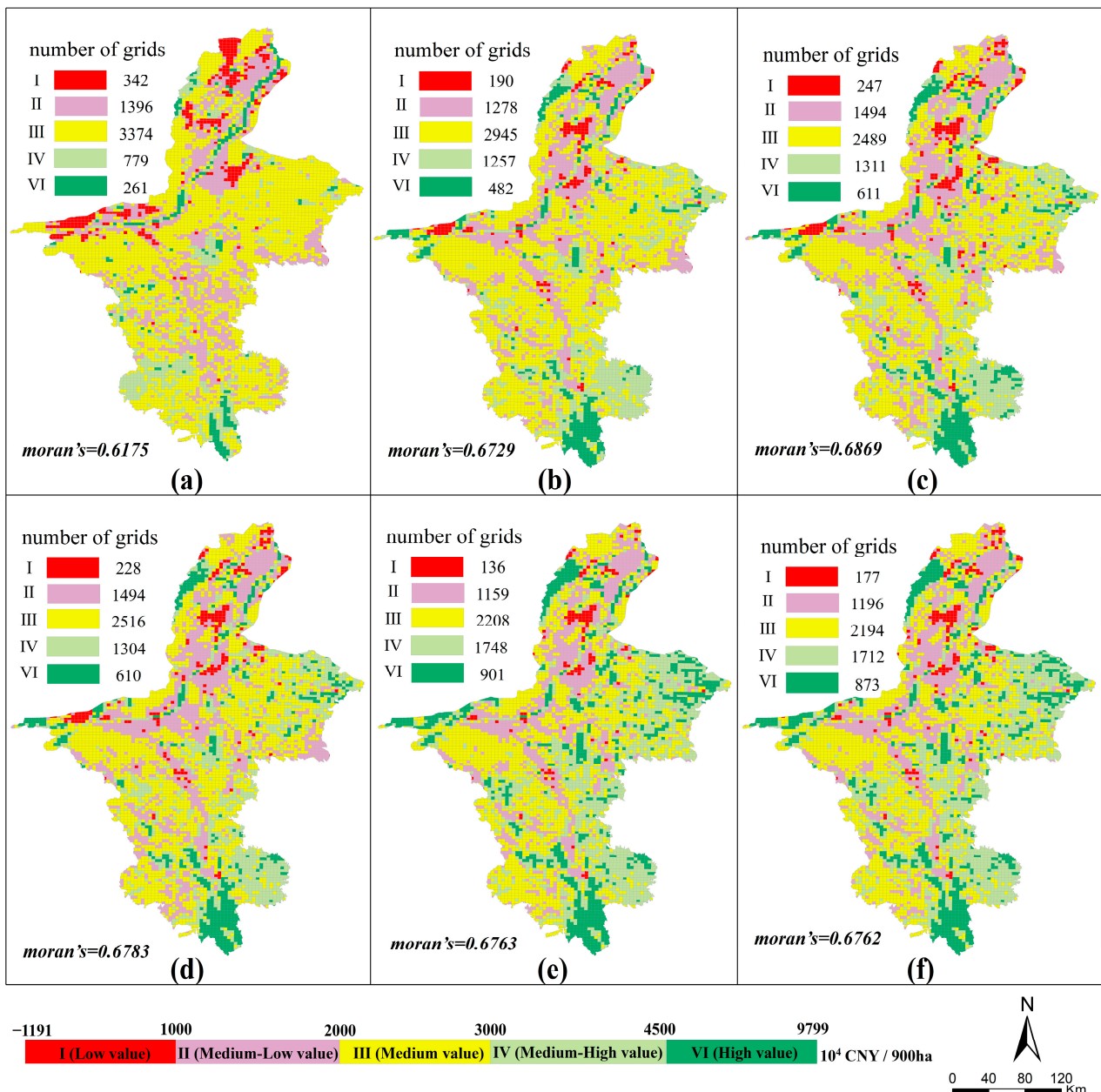

**Figure 7.** The spatial distribution of ESVs: (**a**) 2000; (**b**) 2020; (**c**) ND; (**d**) RED; (**e**) ELP; (**f**) SD.

The spatial distribution of ESVs in the study area shows the characteristic of gradually increasing from north to south and from west to east (Figure 7). The spatial pattern is highly consistent with the land use types. The low value is mainly distributed in and around cities. The medium-low value is mainly distributed in the Ningxia Plain and south-central areas, which are consistent with the distribution of cropland. The medium-high value is mainly distributed in the Yellow River and in woodland and grassland-intensive areas such as the Helan Mountains and Liupan Mountains. The ESV distribution pattern is basically stable in different periods, with significant spatial clustering and increasing agglomeration. Analyzed from the spatial distribution of ESV changes, ESV decreases are mainly distributed in the areas around cities, flat terrain, and frequent human activities. It is noteworthy that the decrease in the ESV along the Yellow River is obvious. The ESV increases mainly in areas with high elevation, abundant rainfall, and low urban sprawl, with significant increases in ecological reserves.

*4.3. Impact of Land Use Changes on the Value of Ecosystem Services*

Based on the equivalence coefficient table (Table S6) and the land use transfer matrix, we calculated the ESV transfer matrix (Figure 8, Table S7) from 2000 to 2030 to analyze the impact of land use quantity change on the ESV. The results show that the increase in woodland is the main type of ESV added. From 2000 to 2020, the increase in woodland increased the ESV by 28.94 billion CNY, contributing 64.24% to the increase in ESV. This contribution rate further increases in 2030, with 79.63%, 93.83%, 93.43%, and 92.57% in ND, RED, ELP, and SD, respectively.

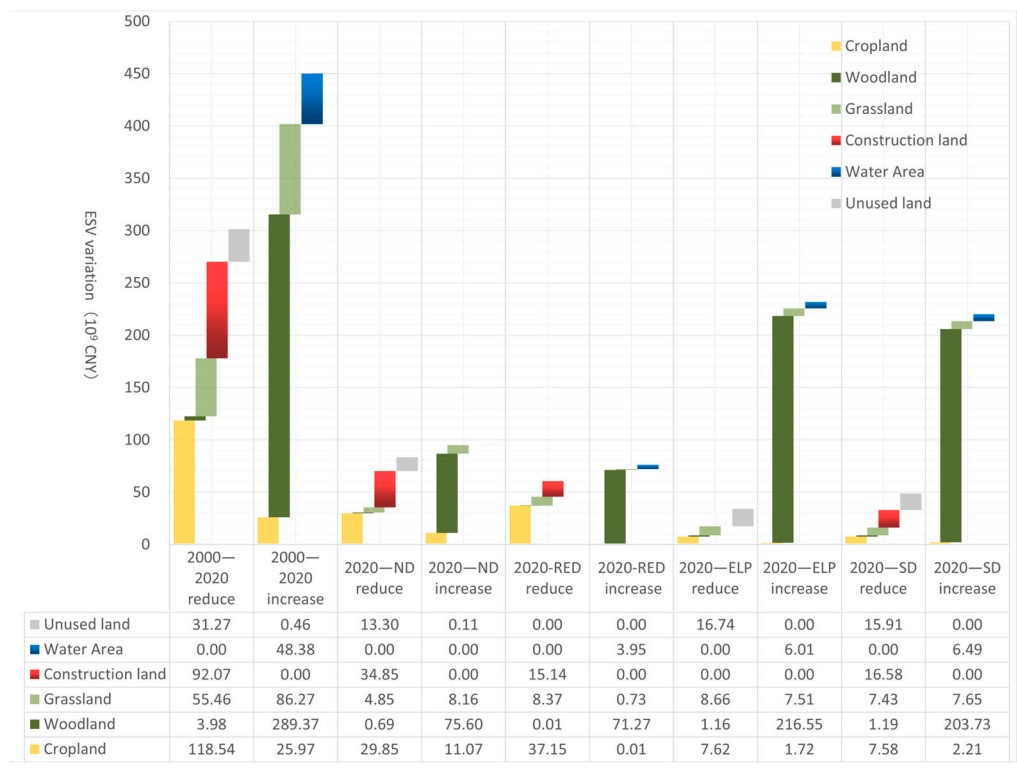

| | 2000—2020 reduce | 2000—2020 increase | 2020—ND reduce | 2020—ND increase | 2020-RED reduce | 2020-RED increase | 2020—ELP reduce | 2020—ELP increase | 2020—SD reduce | 2020—SD increase |
|---|---|---|---|---|---|---|---|---|---|---|
| Unused land | 31.27 | 0.46 | 13.30 | 0.11 | 0.00 | 0.00 | 16.74 | 0.00 | 15.91 | 0.00 |
| Water Area | 0.00 | 48.38 | 0.00 | 0.00 | 0.00 | 3.95 | 0.00 | 6.01 | 0.00 | 6.49 |
| Construction land | 92.07 | 0.00 | 34.85 | 0.00 | 15.14 | 0.00 | 0.00 | 0.00 | 16.58 | 0.00 |
| Grassland | 55.46 | 86.27 | 4.85 | 8.16 | 8.37 | 0.73 | 8.66 | 7.51 | 7.43 | 7.65 |
| Woodland | 3.98 | 289.37 | 0.69 | 75.60 | 0.01 | 71.27 | 1.16 | 216.55 | 1.19 | 203.73 |
| Cropland | 118.54 | 25.97 | 29.85 | 11.07 | 37.15 | 0.01 | 7.62 | 1.72 | 7.58 | 2.21 |

**Figure 8.** Increases and decreases in ESV due to increased land use.

Construction expansion and cropland reclamation are important types of ecological service losses. From 2000 to 2020, the ESV is reduced by 9.21 billion CNY due to construction expansion, accounting for 30.56% of the total ESV loss. Similarly, under the ND scenario, RED scenario, and SD scenario, the ESV was reduced by 3.49, 1.51, and 1.66 billion CNY due to construction expansion, accounting for 41.72%, 24.96%, and 34.05% of the total ESV loss, respectively. Under the rigid constraints of cropland protection, many lands have been reclaimed into cropland, resulting in significant ESV losses. During 2000–2020, ND, RED, ELP, and SD have reduced the ESV by 11.85, 2.99, 3.72, 0.76, and 0.76 billion CNY due to reclaiming farmland, respectively.

## 5. Discussion

*5.1. Significance of the ESV-GMOP-PLUS Model for Future Land Use Optimization*

Achieving sustainable land use through optimal allocation of land resources is an important goal pursued by land managers [57]. The ESV-GMOP-PLUS model we constructed realizes the optimization of the quantitative demand and spatial pattern of land from the perspective of practical management of land, with full consideration of the ecological and economic benefits of land.

The experimental results verify the effectiveness of the model in land planning and ecological protection. The growth rate of construction land slows down considerably in all four scenarios. However, in the ND scenario, the scale of construction land exceeds

the planning control indicators and extends beyond the BUD (Figure 6). This does not happen in any of the scenarios with BUD constraints. During 2000–2020, the proportion of construction land expansion occupying cropland is as high as 43.95%, but in 2030, this proportion drops sharply, and construction land expansion mainly occupies grassland. Meanwhile, all four scenarios saw an increase in farmland. This is mainly because we consider PBC protection and food security in our constraints. It is worth noting that the inter-conversion of land has changed due to a combination of planning constraints and policy orientation (food security and ecological security have become important aspects of national security). During 2000–2020, inter-conversion between different categories occurred. In 2030, the conversion of land types tends to be fixed, with the flow of mainly grassland and unused land to other land types. In addition, we considered the eco-efficiency of the land during land optimization, especially under the ELP and SD scenarios; the woodland with a high ESV has been maintaining a high growth trend, especially the woodland within the RLE (Figure 6).

In summary, the ESV-GMOP-PLUS model can maximize the simulation of land use under planning constraints and identify the optimal land use (i.e., the SD scenario), while meeting the needs of various types of land use and ecological protection. Land use under the ELP and RED scenarios is also provided for decision makers' reference. In addition, the ND scenario can provide a reference for judging the rationality of land planning. Meanwhile, the breakthroughs of PBC and BUD under the ND scenario can be monitored as high-risk areas for illegal land use. The areas of increased ecological land such as forests and grasslands under the SD scenario can be used as a reference for the selection of areas for the implementation of forestation projects.

*5.2. Impact of Land Use Policies on Changes in Ecosystem Services Value*

Land management policies can change the land cover and thus affect the ESV. For example, the policy of "returning cropland to woodland" promotes ESV increase, and policies such as "cropland land requisition-compensation balance" and "linking the increase in land used for urban construction with the decrease in land used for rural construction" impair the ESV. To improve the ecosystem, Ningxia launched the project of returning cropland to woodland in 2000, which includes two components: returning slope cropland to woodland and afforestation on barren hills and wastelands. This promotes the conversion of other land use types to woodland and increases the ESV [58,59]. Although the return of cropland to woodland has reduced food production services, the total ESV has greatly increased. Over the past 20 years, the flow of cropland to woodland and grassland to woodland increased the ESV by 7.54 and 15.48 billion CNY, respectively (Table S7). The increase in woodland contributed to 64.24% of the increase in the ESV. In addition, returning cropland to woodland only gives farmers compensation of 22,500 CNY per hectare, which is equivalent to the value of 10 years of food production. Farmers' interests were not fully protected, forcing them to return woodland back to farming [58], which greatly reduced the ecological benefits of returning cropland to woodland. This assertion can be confirmed by comparing the officially announced area of cropland return to woodland and grass (8967 km$^2$) with the increase in forest land in the land survey (3909 km$^2$). To protect cropland, the Chinese government has implemented policies such as the "cropland land requisition-compensation balance" and "permanent basic cropland protection". Although these policies have ensured food security and increased supply services, a large amount of land has been reclaimed as cropland, resulting in a large reduction in regulating service and support services. From 2000 to 2020, the conversion of woodland, grassland, and water areas to cropland caused ESV losses of 5.40, 3.52, and 2.93 billion CNY, respectively (Table S7). Construction land expansion is another important factor in ESV loss. A total of 9207 million CNY in ESV loss was due to construction from 2000 to 2020 (Table S7), and this trend will continue in 2030.

It is worth noting that the lack of forward-looking and systematic planning of policies is also an important reason for the loss of ESV. Take the development and utilization of

the Yellow River beach land in Ningxia as an example. In the 1990s, the local government encouraged people to develop the riverbank land, resulting in many riverbanks being reclaimed into cropland. There are more than 13,000 hectares of cropland within the Yellow River channel in Ningxia. Although food security was ensured in the short term, it caused degradation to the ecosystem and a reduction in the water conservation function. From 2000 to 2020, 2.9 billion CNY of ESV loss was caused by the conversion of water area into cropland. With the proposed ecological protection and high-quality development of the Yellow River basin, the Yellow Riverbank cropland will again be withdrawn by way of ecological restoration.

*5.3. The Necessity and Urgency of Integrating ESV into Land Management*

Although the Chinese government has placed a high priority on the construction of ecological civilization and has attempted to assess managers through natural resource balance sheets [60], natural resource use quantities (forest coverage rate, etc.), and other methods to achieve an "ecological performance perspective", little consideration has been given to the ESV. This study shows that incorporating the ESV into land management decisions is necessary because it can provide managers with a simple and easily understood single monetary indicator like GDP, allowing them to measure the contribution of ecosystem services to the economy. In Ningxia, the ESV was much higher than GDP in 2000 (GDP was 29.5 billion CNY and the ESV was 138.5 billion CNY). Even though the economy grew rapidly in the following decade and GDP increased 5.3 times, the ESV remained almost equal to GDP (GDP was 157.2 billion CNY and the ESV was 153.3 billion CNY).

It is also urgent to incorporate the ESV into land management decisions as soon as possible. In the study area, the phenomenon of "governance while destruction" in ecological construction is serious. From 2000 to 2020, the ESV increase of 45.05 billion CNY was accompanied by an ESV loss of 30.13 billion CNY (Figure 8). This phenomenon is more severe under the ND scenario and the RED scenario. In the ND scenario, the increase in the ESV of 9.49 billion CNY is accompanied by a decrease of 8.35 billion CNY. In the RED scenario, the increase in the ESV of 7.60 billion CNY is accompanied by a decrease of 6.07 billion CNY. However, in the ELP and EEB scenarios, ESV changes are dominated by increases. Therefore, taking the ESV into full consideration when making land decisions can effectively alleviate the phenomenon of "governance while destruction" and make the integrated development of economic construction and ecological construction safeguard the well-being of human beings.

Although there is still a long way to go to integrate ESVs into land management, it is possible to change the management concept of land managers from an "economic performance concept" to an "ecological performance concept" through lower precision ESV estimation first and then by gradually applying it in land use planning, ecological compensation, and other land management through higher precision ESV estimation.

*5.4. Realism in Land Optimization Simulations*

Firstly, we ensured the authority and authenticity of the foundational data. Land survey data are considered the sole statutory data for land cover, offering a more accurate reflection of land use changes compared to commonly used remote sensing interpretation data. Additionally, our data predominantly originate from government agencies, ensuring the reliability, comprehensiveness, and timeliness of the data. Secondly, our imposed constraints closely align with land management practices. We thoroughly considered constraints related to planning controls, socio-economic conditions, and strategic objectives. Factors such as TSP, PBC, and BUD were incorporated into the spatial constraints. When establishing quantity constraints, we accounted for the government's stringent planning requirements for forests, grasslands, and arable land as well as the specific circumstances in Ningxia. Lastly, our model exhibits high precision. The simulation model we constructed boasts the Fom of 0.2526 and Kappa of 0.7442, ensuring the model's accuracy and the rationality of its parameter settings.

*5.5. Limitations and Future Research Directions*

Our research still has some limitations. First, we incorporate the RLE as a driver rather than a spatial constraint in land optimization. Although it can avoid the disadvantage of fixed ecological constraints that cannot be converted between land use categories, it cannot reflect the rigid control of the RLE. Therefore, in the future, we should explore a REL constraint model that only allows for mutual conversion between woodland, grassland, and unused land. Secondly, the accuracy of land surveys has reached 1 m, and there are 9 major categories and 48 subcategories of land use types. Although we have tried as much as possible to improve the resolution of land optimization and refine the land use types, we are still unable to meet the needs of land management due to limitations in algorithms and computing power. Third, the existing ecological service value coefficients do not cover the types of land use used in land management and fail to adequately take into account the complexity and dynamics of ecosystems. In subsequent studies, the estimation accuracy can be improved by value coefficient correction, the unification of land use types, the subdivision of land use types, and assigning value coefficients consistent with their ecological function.

**6. Conclusions**

We proposed a model coupling ESV, GMOP, and PLUS, used land survey data, and added TSP constraints to simulate the land use structure and ESV of Ningxia in 2030 under the ND, RED, ELP, and SD scenarios. We found that (1) TSP played an important role in stopping the uncontrolled expansion of construction land, improving the effectiveness of ecological land use, and promoting food security; (2) in the future, construction land, woodland, and cropland will be the main expansion land categories, while grassland and unused land, which lack strict use control, will be the main types of land outflow; (3) although the total ESV has increased steadily and slightly, the spatial distribution has become more and more concentrated. However, the phenomenon of "governance while destruction" in ecological construction is serious, which is alleviated under the ELP scenario and SD scenario; (4) we offered a wide range of possibilities for future land use. The ND scenario has the highest rate of construction land expansion (18.57%) and the lowest ESV (153.75 billion CNY). The RED scenario has the highest land economic efficiency and the largest amount of ecological land reduction (−0.77%). The ELP scenario has the highest ESV (173.12 billion CNY). The SD may be more suitable for future regional development, although the ESV (17.51 billion CNY) is slightly lower than ELP, but the economic benefits have greatly improved. Our study provided a reference for an early warning of land risks and sustainable development. Land modeling based on land survey and management data can facilitate academic research in practical management applications.

**Supplementary Materials:** The following supporting information can be downloaded at: https://www.mdpi.com/article/10.3390/land13040557/s1, Table S1: Comparison of remotely interpreted land-use data with land survey data; Table S2: Categories of land use; Table S3: Parameters of ELAS under the four scenarios in Ningxia; Table S4: Parameters of cost matrix under the four scenarios (ND/RED/ELP/SD) in Ningxia; Figure S1: Comparison of simulated and actual LULC in 2020; Table S5: The equivalent coefficients of ecosystem service provided by per unit area of terrestrial ecosystems in China; Table S6: Ecological service value per unit area of different terrestrial ecosystems in Ningxia; Table S7: Land use and ESV transfer matrix of Ningxia from 2000 to 2030.

**Author Contributions:** Conceptualization, R.S. and Z.W.; methodology, R.S.; software, N.G. and K.H.; validation, R.S., M.W. and Z.W.; investigation, N.G.; resources, N.G.; data curation, R.S. and Y.Z.; writing—original draft preparation, R.S. and N.G.; writing—review and editing, R.S. and Z.W.; visualization, R.S. and Y.Z.; supervision, Z.W. All authors have read and agreed to the published version of the manuscript.

**Funding:** This research was funded by the Key Project of the National Social Science Foundation of China (No. 23AZD058), the Natural Science Foundation of Ningxia Province (No. 2021AAC03060), and the Ningxia Young Talent Support Project.

**Data Availability Statement:** The authors can provide the data upon reasonable request. The data are not publicly available due to privacy restrictions.

**Acknowledgments:** The authors are particularly grateful to all researchers for providing data support for this study.

**Conflicts of Interest:** The authors declare no conflicts of interest.

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
