# Peer review of "Multi-Scenario Land Use Optimization Simulation and Ecosystem Service Value Estimation Based on Fine-Scale Land Survey Data"

_land, doi:10.3390/land13040557_

Round 1

Reviewer 1 Report

Comments and Suggestions for Authors

In the manuscript titled Multi-scenario Land Use Optimization Simulation and Ecosystem Service Value estimation based on fine-scale land survey data, used land survey data to coupling ESV, Grey multi-objective optimization and patch generation land use simulation models based on authoritative land management data to land use and ESV changes under natural development, rapid economic development, land ecological protection and sustainable development scenarios to 2030,fully considered the land planning control of land use.This research provides a new idea for optimizing land allocation and improving land resource utilization efficiency in the future. It is an innovative research paper, but there are some formatting problems that need to be adjusted.

1.        The text annotation in the image should try not to overlap with the image.

2.        The red legend in Figure 7 is not clearly marked; You can use the same layout as in Figure 6, placing the north pointer above the scale and lengthening each legend a bit.

3.        It is suggested to add axes and scale lines to Figure 8, which may be more clear.

Comments on the Quality of English Language

English needs further improvement

Reviewer 2 Report

Comments and Suggestions for Authors

The manuscript is well organized and the information is well presented. The methods are appropriate. I have two suggestions:

In recent years, the concept of ecosystem services has evolved to include nature's contributions to human well-being. I suggest including some statements in the introduction (end of the first paragraph?) as well as in the discussion (implying a change in your results?).

When land use modeling is done, it is difficult to communicate to decision makers the veracity of the results. Also, how they can be applied. My suggestion to the authors is to emphasize these two points, perhaps towards the end of the discussion and in the conclusions.

Congrats,

Reviewer 3 Report

Comments and Suggestions for Authors

Thank you for submitting your paper to Land Journal. We appreciate the effort you've put into your work, and we're pleased to inform you that overall, it appears to be of high quality and well-presented. However, we have identified a few minor points that need attention before we can proceed with publication:

1. Please ensure that the resolution of the images in Figures meets the journal's standards.

2. Please add a vertical title to Figure 8 for clarity.

3. In Table 2, the water area shows an increasing trend in the rapid economic development (RED) scenario. While it's clear why there's an increase in the ecological land protection (ELP) and sustainable development (SD) scenarios, please provide an explanation for the increasing trend in the RED scenario as well. You may use the data presented in Table 2 to elaborate on this reason.
